# An Oxidation Study of Phthalimide-Derived Hydroxylactams

**DOI:** 10.3390/molecules27020548

**Published:** 2022-01-15

**Authors:** Bernard L. Adjei, Frederick A. Luzzio

**Affiliations:** Department of Chemistry, University of Louisville, 2320 South Brook Street, Louisville, KY 40292, USA; bernard.adjei@louisville.edu

**Keywords:** hydroxylactams, IBX, imides, isoindolin-1-ones, NiO_2_, oxidation, PCC, phthalimides

## Abstract

A systematic study of the oxidation of 3-hydroxy-2-substituted isoindolin-1-ones (hydroxylactams) and their conversion to the corresponding phthalimides was undertaken using three oxidants. Of special interest was the introduction of nickel peroxide (NiO_2_) as an oxidation system for hydroxylactams and comparison of its performance with the commonly used pyridinium chlorochromate (PCC) and iodoxybenzoic acid (IBX) reagents. Using a range of hydroxylactams, optimal conversions of these substrates to the corresponding imides was achieved with 50 equivalents of freshly prepared NiO_2_ in refluxing toluene over 5–32 h reaction times. By comparison, oxidations of the same substrates using PCC/silica gel (three equivalents) and IBX (three equivalents) required oxidation times of 1–3 h for full conversion but required lengthier purification. While nominal amounts (~25 mg) of substrate hydroxylactams were used to ascertain conversion, scale-up procedures using all three methods gave good to excellent isolated yields of imides.

## 1. Introduction

The *N*-phthaloyl (phthalimide) group is a well-established, frequently used protecting group for amine functionality in organic synthesis, and apart from its employment as a protecting group (2-substituted-l-isoindoline-1,3-dione group), it is used as a fundamental means for introducing nitrogen (Gabriel reaction) in a wide range of starting materials and intermediates in organic synthesis [1]. The unique serviceability of the *N*-phthaloyl group is enhanced by its mild removal from compatible intermediates by a brief treatment with alcoholic hydrazine thereby releasing the desired amine functionality [2]. The *N*-phthaloyl group **I** (Figure 1) can also serve as a starting point for isoindolinones (lactams) **II** through its deoxygenation and deoxygenation of its partial reduction product, the corresponding hydroxylactam **III** [3]. The partial reduction of the imide to the hydroxylactam can be facile and, in some cases, can interfere with the phthaloyl group’s efficacy as a protecting group in a synthetic scheme. The interchange between the phthaloyl group and the isoindolinone group can be used to advantage, whereby the isoindolinone (lactam, **II**) group can act as a latent protecting group by oxidation back to the imide, which can then be easily removed [4]. The importance of hydroxylactams is multifold, as these compounds are well-established intermediates toward molecules of greater complexity through acyliminium ion chemistry [5], and their value in the synthesis of medicinally important compounds has been demonstrated [6]. The reduction of cyclic imides to hydroxylactams is routinely performed with sodium borohydride in alcoholic media, although the less commonly used aluminum amalgam [Al(Hg)] [7] and diisobutylaluminum hydride (DIBAL) can be employed [8], with Al(Hg) being selective for reduction of the benzylic carbonyls of phthalimides. As mentioned above, inadvertent reduction of a phthalimide protecting group to a hydroxylactam may accompany a desired reduction of a separate functional group within the same molecule. When inadvertent reduction of the imide occurs, a useful transformation would entail the oxidation of the hydroxylactam back to the imide, so its removal may be facilitated. In this report, we detail the facile oxidation of phthalimide-derived hydroxylactams (**1a–j**) to the corresponding phthalimides (**2a–j**) using a nickel peroxide (NiO_2_) oxidation system [9] (Figure 1). Included in the study is the procurement of the starting imides (from the starting amines) and the hydroxylactams (through sodium borohydride reduction). Both the imides and the hydroxylactam substrates are non-commercially available but have been previously reported. In terms of contrast and comparison, the title oxidative conversions are also accomplished with the common alcohol oxidants pyridinium chlorochromate/silica gel (PCC/SiO_2_) [10] and the hypervalent iodine reagent *o*-iodoxybenzoic acid (IBX) [11,12,13]. While the main thrust of the NiO_2_/PCC/IBX oxidations detailed herein was to ascertain only the conversion of the hydroxylactams to the imides over time, scale-up procedures using all three oxidants on a single selected starting material were examined.

## 2. Results and Discussion

We began the study with the employment of the nickel peroxide (NiO_2_) oxidant, which was prepared by the method of Nakagawa [9], and for our purposes, no characterization of the oxidant described therein was necessary. The preparation of NiO_2_ essentially entailed the treatment of nickel sulfate hexahydrate (Ni_2_SO_4_·6H_2_O) with commercial bleach (sodium hypochlorite) in the presence of aqueous sodium hydroxide. The resultant finely divided black precipitate was collected by filtration and scrupulously dried at room temperature before its use in the oxidations. While nickel peroxide is commercially available (USD 138.00/5 g), we found that our laboratory preparation is markedly less expensive (USD 7.50/g) and, in our hands, gave reproducible results since the shelf life is known. Moreover, due to the same issue of the comparative inexpensiveness and reproducibility of freshly prepared NiO_2_, we deemed it unnecessary to try to reclaim or otherwise reuse it. The hydroxylactam substrates **1a–j** (Table 1) were prepared by a fairly standard reduction of the corresponding phthalimides **2a–j** using sodium borohydride. The physical properties of the substrate compounds **1a-j** were consistent with literature values (see Experimental Section (Section 3)). While the phthalimides **2a–j** were relatively simple starting materials, they were not commercially available and were mainly prepared by the method of Pietka-Ottlick [14]. Since phthalimides **2a–j** were used as both starting materials and to corroborate the formation of the NiO_2_ oxidation products, their identity was confirmed by properties previously reported in the literature. Prior to evaluating NiO_2_ on the range of substrates **1a–j** detailed in Table 1, 3-hydroxy-2-propylisoindolin-1-one **1a** (see Table 1) was used as a test substrate for solvent efficiency. The solvent systems evaluated using **1a** were dichloromethane (DCM), 1,2-dichloroethane (DCE), pyridine and toluene. No conversion to the corresponding imide **2a** was observed for the lower-boiling chlorinated solvents (DCM, DCE), even during extended reaction times. Pyridine (reflux) did mediate some conversion over a 24-h reaction period; however, toluene was far superior, thus allowing both complete conversion and the ease of purification of the product by comparison. The NiO_2_-mediated oxidations detailed in Table 1 utilized toluene as a reaction solvent and were run on a 25 mg scale of substrate. 2-*n*-Hexyl-3-hydroxyisoindolin-1-one **1d**, and 2-(4-fluorophenyl)-3-hydroxy isoindolin-1-one **1f** exhibited the shortest conversion times (5 h) to the corresponding phthalimides **2d** and **2f**, while the NiO_2_-mediated oxidation of substrates **1a**, **1h** and **1j** took 24 h to complete. The PCC/silica gel oxidations of **1a**–**j** were conducted according to the protocol developed in these laboratories some time ago and utilized commercially available shelf-stable PCC ground with silica gel (60–230 mesh). The yellow-orange powder was combined with dichloromethane to form the oxidizing mixture. The substrates were then added, and the reactions were conducted at room temperature and monitored periodically by thin-layer chromatography. Overall, the shortest conversion times were observed with PCC/silica and ranged from 2.5 to 12 h, with the average conversion time for the ten substrates calculated to be 6.5 h. The IBX-mediated oxidations were comparable to those mediated by PCC/SiO_2_, with the exception being the *n*-hexyl substrate **1d**, whereby the elapsed reaction time was 22 h versus the 12 h required for the PCC/SiO_2_.

Although the oxidations using NiO_2_, PCC/SiO_2_ and IBX proceed through well-known but distinctly different mechanisms, no structural trends could be observed from using the diversity of substrates shown in Table 1. The reactions denoted in Table 1 were for strictly elapsed time information; they were not worked up, and the products were not isolated. However, as a matter of practical application, scale-up reactions using NiO_2_ were performed on 3-hydroxy-2-phenethylisoindolin-1-one (**1g**) on a scale of 300, 600 and 900 mg, and the isolated yields of the product 2-phenethylisoindoline-1,3-dione **2g** were 62, 75 and 64%, respectively, after purification by recrystallization (see Experimental Section (Section 3)). The same series of scale-up experiments (300, 600, 900 mg) using substrate **1g** was performed with IBX (three equivalents) in refluxing acetonitrile and provided isolated yields of **2g** in quantitative yields for all three examples. Similar to the scale-up examples performed with NiO_2_, the pure product **2g** was obtained by recrystallization. Historically, the main issue with PCC and other oxochromium (VI)-amine oxidants is the management and removal of the polymeric reduced chromium by-products that oftentimes complicate the purification of the desired products. In addition, the greater the scale-up with PCC, the more difficult the management of the reduced chromium by-products. The scale-up oxidation with PCC/silica using a representative substrate **1g** demonstrated the ease of purification allowed by using the oxidant/adsorbent mixture (see Experimental Section (Section 3)) and resulted in 83, 76 and 69% isolated yields on 900, 600 and 300 mg scales, respectively. Similar to the PCC/SiO_2_ oxidations, the IBX-mediated oxidations utilized 0.8–0.11 mmole (25 mg) of substrate and three equivalents of IBX (Table 1). The solvent of choice was refluxing acetonitrile (80 °C/oil bath). As compared with the PCC/SiO_2_-mediated oxidations, workup and purification were greatly simplified despite the slightly slower reaction times. 

An interesting case in point involved the oxidation of the hydroxylactam **3** derived from the 5-phthalimido-2,3-isopropylidene-β-O-methylriboside **4**. Reduction of the known riboside **4 [15,16]** with sodium borohydride gave the novel hydroxylactam **3** as a diastereomeric mixture in quantitative yield (Figure 2). 

On a 25 mg scale, treatment of **3** with nickel peroxide (50 equivalents) in refluxing toluene resulted in the formation of product **4** within an hour and, after 12 h, gave an 81% isolated yield of imide **4** (Figure 2). By comparison, oxidation of **3** with PCC/SiO_2_ (three equivalents) in dichloromethane afforded a 60% isolated yield of the ribosyl imide **4** after a reaction time of 8 h. Finally, the IBX-mediated oxidation of **3** provided a 75% isolated yield of ribosyl imide **4** after a reaction time of 24 h using refluxing acetonitrile (80 °C) as a reaction medium. We will note that normally PCC and oxochromium (VI) reagents in general are not the oxidants of choice for hydroxyl groups borne by carbohydrate scaffolds, so consequently, the hypervalent iodine oxidants have steadfastly offered a superior alternative. As with any oxidation system utilizing a metal-derived or organic-based oxidant, the option of adapting or converting the procedure to a catalytic version is always an interesting option. We were aware of previous work involving catalytic NiO_2_ systems used to oxidize simple alcohols to carbonyl compounds such as benzyl alcohol to benzoic acid or phenethyl alcohol to acetophenone [17,18]. While the catalytic systems are of great merit when used with the simpler substrates, the highly acidic and relatively basic nature of these systems precludes their use with acid-labile substrates such as 3, as well as the base-sensitive phthalimide moiety. Hence, the conditions described in the previously described catalytic systems precluded their application to our substrates. 

## 3. Experimental Section

### 3.1. General Methods

Solvents and reagents are ACS grade and were used as commercially supplied with the exception of hexane. ACS-grade hexane was purified for chromatography according to the method described by Perrin [19]. Tetrahydrofuran (THF) was distilled under nitrogen from a mixture of sodium/benzophenone. Analytical thin-layer chromatography (TLC) utilized 0.25 mm pre-cut glass-backed plates (Silica Gel 60 F254, EMD Millipore, Burlington, MA, USA). Thin-layer chromatograms were visualized during chromatographic extraction and reaction runs by rapidly dipping the plates in anisaldehyde/ethanol/sulfuric acid stain or phosphomolybdic acid/ethanol stain and heating (hot plate). Carbohydrate intermediates and products were visualized on TLC by rapidly dipping the plate into 10% sulfuric acid/ethanol and charring (hot plate). Flash-column chromatography utilized silica gel 60 (230–400 mesh, E. Merck, Darmstadt, Germany). Melting points were taken on a Mel-Temp apparatus. Extracts and chromatographic fractions were concentrated with a Büchi rotavapor under water aspirator vacuum. Disposal of chromium waste was performed in accordance with the National Research Council publication “Prudent Practices in the Laboratory” [20]. Nuclear magnetic resonance (^1^H and ^l3^C NMR) spectra were recorded at 400 and 100 MHz respectively with a Varian VNMRS 400 (Agilent Technologies, Santa Clara, CA, USA), and at 500 and 125 MHz respectively with a AS 500 MHz instrument (Oxford Instruments, Beverly MA, USA) using CDCl_3_ (with TMS as internal standard) and DMSO as NMR solvents. Infrared spectra (Fourier Transform Infrared Spectroscopy, FTIR) were recorded with a Spectrum 100 instrument (Perkin-Elmer, Waltham, MA, USA). HRMS were measured at Indiana University Mass Spectrometry Facility.

### 3.2. Preparation of the Phthalimides **2a–j**

Phthalimides **2a–j** were prepared by the method of Pietka-Ottlick [13], and the analytical information for these compounds along with the associated references is detailed below. 

*2-Propylisoindoline-1,3-dione* **2a**: White crystals from ethanol: m.p. 63–65 °C (Lit. [21] 65–67 °C); R_f_: 0.5 (hexane/ethyl acetate, 2:1).

*2-Phenylisoindoline-1,3-dione* **2b**: White crystals from ethanol: m.p. 208–210 °C (Lit. [22] 209–210 °C); R_f_: 0.45 (hexane/ethyl acetate, 2:1).

*2-Cyclohexylisoindoline-1,3-dione* **2c**: White crystals from ethanol: m.p. 169–171 °C (Lit. [23] 169–172 °C); R_f_: 0.45 (hexane/ethyl acetate, 2:1).

*2-n-Hexylisoindoline-1,3-dione* **2d**: Low-melting solid from ethanol: the ^1^H NMR is consistent with that previously reported [24]; R_f_: 0.6 (hexane/ethyl acetate, 2:1).

*2-Benzylisoindoline-1,3-dione* **2e**: White crystals from ethanol: m.p. 114–116 °C (Lit. [25] 112–113 °C); R_f_: 0.47 (hexane/ethyl acetate, 2:1).

*2-(4-Fluorophenyl)isoindoline-1,3-dione* **2f**: White crystals from ethanol: m.p. 181–183 °C (Lit. [26] 179–181 °C); R_f_: 0.40 (hexane/ethyl acetate, 2:1).

2-Phenethylisoindoline-1,3-dione **2g**: White crystals from ethanol: m.p. 129–131 °C (Lit. [27] 129–131 °C); R_f_: 0.50 (hexane/ethyl acetate, 2:1).

*2-(4-Methoxyphenyl)isoindoline-1,3-dione* **2h**: Light yellow crystals from ethanol: m.p. 161–163 °C (Lit. [28] 160–162 °C); R_f_: 0.36 (hexane/ethyl acetate, 2:1).

*(4-Methoxyphenethyl)isoindoline-1,3-dione* **2i**: White crystals from ethanol: m.p. 135–138 °C (Lit. [29] 135–138 °C); R_f_: 0.40 (hexane/ethyl acetate, 2:1). 

2-(4-Bromophenyl)isoindoline-1,3-dione **2j**: White crystals from ethanol: m.p. 205–207 °C (Lit. [22] 205–206 °C); R_f_: 0.47 (hexane/ethyl acetate, 2:1).

### 3.3. Preparation of Hydroxylactams (**1a-j**) from Imides **2a–j**

The preparation of the hydroxylactams **1a-j** from the imides **2a-j** utilized the general method of Castro-Castillo [6], and the analytical details of the products along with the pertinent references are detailed below.

*3-Hydroxy-2-propylisoindolin-1-one* (**1a**): White crystals from methanol/water: m.p. 89–91 °C (Lit. [5] 92–93 °C; R_f_ 0.2 (hexane/ethyl acetate, 2:1).

*3-Hydroxy-2-phenylisoindolin-1-one* (**1b**): White crystals from methanol/water: m.p. 167–169 °C (Lit. [30] 167–168 °C; R_f_ 0.3 (hexane/ethyl acetate, 2:1).

*2-Cyclohexyl-3-hydroxyisoindolin-1-one* (**1c**): White crystals from methanol/water: m.p. 124–126 °C (Lit. [7] 126–128 °C); R_f_ 0.17 (hexane/ethyl acetate, 2:1).

2-n-Hexyl-3-hydroxyisoindolin-1-one (**1d**): Off-white crystals from methanol/water: m.p. 121–123 °C; the ^1^H NMR is consistent with that previously reported [31]; R_f_ 0.17 (hexane/ethyl acetate, 2:1).

*2-Benzyl-3-hydroxyisoindolin-1-one* (**1e**): White crystals from methanol/water: m.p. 141–143 °C (Lit. [32] 145–146 °C); R_f_ 0.19 (hexane/ethyl acetate, 2:1). 

*2-(4-Fluorophenyl)-3-hydroxyisoindolin-1-one* (**1f**): White crystals from methanol/water: m.p. 189–191 °C (Lit. [33] 190–192 °C); R_f_ 0.20 (hexane/ethyl acetate, 2:1).

*3-Hydroxy-2-phenethylisoindolin-1-one* (**1g**): White crystals from methanol/water: m.p. 161–163 °C (Lit. [6] 170–172 °C); R_f_ 0.20 (hexane/ethyl acetate, 2:1).

3-Hydroxy-2-(4-methoxyphenyl)isoindolin-1-one (**1h**): Off-white crystals from methanol/water: m.p. 155–157 °C (Lit. [34] 156–157 °C); R_f_ 0.20 (hexane/ethyl acetate, 2:1).

*3-Hydroxy-2-(4-methoxyphenethyl)isoindolin-1-one* (**1i**): White crystals from methanol/water: m.p. 117–119 °C (Lit. [35] 113–114 °C); R_f_ 0.10 (hexane/ethyl acetate, 2:1).

*2-(4-Bromophenyl)-3-hydroxyisoindolin-1-one* (**1j**): White crystals from methanol/water: m.p. 174–176 °C (Lit. m.p. not reported [36]); R_f_ 0.10 (hexane/ethyl acetate, 2:1). The corresponding 2-(4-bromophenyl)-3-methoxyoxyisoindolin-1-one was prepared from **1j**, and its ^1^H NMR spectra were comparable with that found in the literature [37].

### 3.4. NiO_2_-Mediated Oxidation Runs and Typical Procedure: Oxidation of 3-Hydroxy-2-phenethylisoindolin-1-one 1g

To a 25 mL round-bottom flask fitted with a magnetic stir bar, 3-hydroxy-2-phenethylisoindolin-1-one **1g** (25 mg, 0.1 mmol) and toluene (10 mL) were added. To the stirred suspension, nickel peroxide (0.45 g, 5 mmol) was added in one portion, and the resultant black suspension was refluxed (oil bath) under a nitrogen atmosphere. The progress of the reaction was monitored by TLC (hexane/ethyl acetate, 2:1) at one-hour intervals, whereby conversion to the product phthalimide **2g** was evidenced by the formation of a more mobile spot. The time was recorded for complete disappearance of the starting material.

### 3.5. PCC/Silica-Mediated Oxidation Runs and Typical Procedure: Oxidation of 3-Hydroxy-2-phenethylisoindolin-1-one 1g

To a 25 mL round-bottom flask fitted with a magnetic stir bar, a mixture of PCC (0.13 g, 0.59 mmol) and silica gel (1.0 wt equivalents), which was ground together in a mortar DCM (10 mL), was added, and the light yellow-orange suspension was stirred at room temperature. 3-Hydroxy-2-phenethylisoindolin-1-one **1g** (25 mg, 0.19 mmol) was added in one portion while stirring under a nitrogen atmosphere. The progress of the reaction was monitored by TLC (hexane/EtOAc, 2:1) at one-hour intervals (see Table 1) until the starting hydroxylactam was consumed.

### 3.6. IBX-Mediated Oxidation Runs-Typical Procedure: Oxidation of 3-Hydroxy-2-phenethylisoindolin-1-one 1g

To a 25 mL round-bottom flask fitted with a stir bar was added 3-hydroxy-2-phenethyl-isoindolin-1-one **1g** (25 mg, 0.19 mmol), followed by acetonitrile (5.0 mL). 2-Iodoxybenzoic acid was added in one portion while stirring, and the resultant white suspension was warmed to 80 °C. The reaction mixture was monitored by TLC (hexane/EtoAc, 2:1) at one-hour intervals until the substrate was consumed and full conversion to product was observed. Upon completion of the reaction, the reaction mixture was cooled to room temperature and vacuum filtered using a sintered glass funnel packed with a bed of Celite. The filter cake was washed with EtOAc (3 × 5 mL), and the combined filtrate was concentrated on a rotary evaporator. The crude product was triturated with hexanes and ethyl acetate and allowed to cool in a refrigerator. Excess solvent was decanted and concentrated on a rotary evaporator to afford the pure product as an off-white crystalline solid. The off-white solid may be vacuum filtered with cold methanol rinse to obtain a white crystalline solid (optional).

### 3.7. NiO_2_-Mediated Scale-Up Oxidation of **1g**: 2-Phenethylisoindoline-1,3-dione **2g**

To a round-bottom flask fitted with a magnetic stir bar, **1g** (900 mg, 3.5 mmol) and toluene (60 mL) were added while stirring. Nickel peroxide (16.1 g, 177 mmol) was added in one portion, and the reaction was refluxed (oil bath) under a nitrogen atmosphere. The progress of the reaction was monitored by TLC (hexane/ethyl acetate, 2:1). Upon completion of the reaction as indicated by TLC, the reaction mixture was cooled to room temperature and vacuum filtered using a sintered glass funnel packed with Celite. The filter cake was washed with EtOAc (3 × 10 mL), and the combined filtrate then concentrated on a rotary evaporator. The crude product was triturated with hexane/ethyl acetate and allowed to cool in a refrigerator. The excess solvent was decanted and the residual solution concentrated to afford 2-phenethylisoindoline-1,3-dione **2g** (0.57 g, 64%) as an off-white crystalline solid. A 300 mg (1.1 mmol) oxidation of **1g** using NiO_2_ gave 0.2 g (67%) of **2g**, while a 600 mg (2.2 mmol) oxidation under the same condition gave 0.45 g (75%) of **2g**. 

### 3.8. PCC/SiO_2_-Mediated Scale-Up Oxidation of **1g**: 2-Phenethylisoindoline-1,3-dione **2g**

A mixture of PCC (2.29 g, 10.65 mmol) and silica gel (70–230 mesh, 2.29 g) was ground together using a small mortar and the yellow-orange powder placed in a 50 mL round-bottom flask fitted with a magnetic stir bar. Dichloromethane (25 mL) was added, and to the yellow-orange suspension was added 3-hydroxy-2-phenethylisoindolin-1-one **1g** (900 mg, 3.553 mmol) in one portion while stirring at 25 °C. The progress of the reaction was monitored by TLC (hexane/ethyl acetate, 2:1), whereby the reaction was complete after four hours. The dark-brown suspension was diluted with ethyl acetate, which precipitated the dark-brown reduced chromium products. The suspension was then vacuum filtered through a Buchner funnel (60 × 50 mm) packed with Celite, and the granular brown residue was washed with diethyl ether (100 mL). The filtrate was concentrated under vacuum to give a light-brown solid, which was purified by recrystallization from hot methanol/water to afford **2g** (0.75 g, 83%) as a white crystalline solid. A 600 mg oxidation of **1g** using the above procedure gave 0.45 of (76%) of **2g**, while a 300 mg oxidation of **1g** under the same conditions gave 0.2 g (69%) of **2g**. 

### 3.9. IBX-Mediated Scale-Up Oxidation of 1g: 2-Phenethylisoindoline-1,3-dione 2g

To a 50 mL round-bottom flask fitted with a magnetic stir bar, 3-hydroxy-2-phenethylisoindolin-1-one **1g** (900 mg, 3.55 mmol) and acetonitrile (20 mL) were added. To the stirred suspension was stirred 2-iodoxybenzoic acid (2.98 g, 10.66 mmol) in one portion. The reaction mixture was then heated (oil bath) at 80 °C for 10 h. The progress of the reaction was monitored by TLC (hexane/ethyl acetate, 2:1). Upon complete consumption of the hydroxylactam, the reaction mixture was cooled to room temperature and vacuum filtered using a sintered glass funnel. The filter cake was then washed with ethyl acetate (3 × 5 mL) and the filtrate concentrated under vacuum. The crude product was triturated with dichloromethane/pentane and cooled in a refrigerator. The excess solvent was then decanted and concentrated under vacuum to obtain 2-phenethylisoindoline-1,3-dione **2g** (0.89 g, 99%) as a white crystalline solid. On a 300 mg and 600 mg scale using substrate **1g**, and employing a reaction time of 8 h, the yields of **2g** were quantitative in both runs.

### 3.10. 2-(((4R,6R)-6-Methoxy-2,2-dimethyltetrahydrofuro [3,4-d][1,3]dioxol-4-yl)methyl) isoindoline-1,3-dione **4**

The 5-*N*-phthalimido riboside **4** was prepared by the combined methods of Ohrui and Kim [15,16], with the exception that the final product was purified by flash-column chromatography (benzene/ethyl acetate, 3:1): m.p. 107–109 °C; Lit. [15] 128 °C.

### 3.11. Sodium Borohydride Reduction of Imide **4**: 3-Hydroxy-2-(((4r,6r)-6-methoxy-2,2- dimethyltetrahydrofuro[3,4-d][1,3]dioxol-4-yl)methyl)isoindolin-1-one (hydroxylactam 3)

To a round-bottom flask fitted with a magnetic stir bar, (((4*R*,6*R*)-6-methoxy-2,2-dimethyltetrahydrofuro[3,4-*d*][1,3]dioxol-4-yl)methyl)isoindoline-1,3-dione (25 mg, 0.075 mmol) was suspended in methanol (2.5 mL) and at 0 °C (ice water bath). Sodium borohydride (8.5 mg, 0.225 mmol) was added portionwise over 2 min, and the mixture was stirred for 1 h. The progress of the reaction was monitored by thin-layer chromatography (toluene/ethyl acetate, 1:1). Upon completion of the reaction, 125 mg of silica gel was added, then the suspension was filtered while washing with methanol. The filtrate was extracted with dichloromethane (3 × 5 mL), washed with water, then brine. The organic layer was dried over Na_2_SO_4_ and concentrated under vacuum to afford 3-hydroxy-2-(((4*R*,6*R*)-6-methoxy-2,2-dimethyltetrahydrofuro[3,4-*d*][1,3]dioxol-4-yl) methyl) isoindolin-1-one **3** as an off-white solid (17 mg, 68%), m.p. 119–121 °C, *R*_f_ 0.37 (toluene/ethyl acetate, 1:1). ^1^H-NMR (400 MHz, CDCl_3_) δ: 7.69–7.63 (m, 2H), 7.52 (m, 4H), 7.44 (m, 2H), 5.95 (s, 1H), 5.84 (s, 1H), 4.95 (s, 2H), 4.74 (dd, 2H, J = 5.2, 13.6 Hz), 4.63 (d, 2H, J = 5.6 Hz), 4.49 (brt, 2H), 4.34 (brt 2H), 3.81 (m, 2H), 3.38 (s, 3H), 3.21 (s, 3H), 1.46 (s, 3H), 1.41 (s, 3H), 1.29 (s, 3H), 1.25 (s, 3H). ^13^C-NMR (100MHz, CDCl_3_) δ: 167.92, 167.76, 144.14, 144.02, 132.53, 132.43, 131.35, 131.22, 129.78 (overlap), 123.44, 123.37, 112.69, 112.63, 110.12, 109.97, 109.87, 85.59, 85.38, 85.28, 85.03, 83.21, 82.39 (overlap), 81.65, 81.53, 55.40, 53.56, 44.23, 41.98, 26.52 (overlap), 25.04 (overlap). FTIR (neat): 1094, 1372, 1423, 1686, 2930, 3332, 3536 cm^−1^; HRMS (ESI) Calcd. For C_17_H_21_NO_5_, 358.1261[M + Na]+ Found: 358.1263.

### 3.12. Oxidation of 3-Hydroxy-2-(((4R,6R)-6-Methoxy-2,2-dimethyltetrahydrofuro [3,4-d][1,3]dioxo l-4-yl)methyl)isoindolin-1-one ***3*** with NiO_2_: Ribosyl Imide ***4***

To a solution of 3-hydroxy-2-(((4*R*,6*R*)-6-methoxy-2,2- dimethyltetrahydrofuro[3,4-*d*][1,3]dioxol-4-yl)methyl)isoindolin-1-one **3** (17 mg) and toluene (2.5 mL) in a 25 mL round-bottom flask fitted with a magnetic stir bar was added NiO_2_ (23 mg, 2.5 mmol) in one portion, and the black suspension was heated at reflux under an N_2_ atmosphere. The progress of the reaction was monitored by TLC (toluene/ethyl acetate, 1:1). Upon full conversion of starting material to product (12 h), the reaction mixture was filtered through a bed of Celite^®^ while washing with ethyl acetate (3 × 5 mL). The filtrate was concentrated under vacuum and the crude product triturated with hexane/ethyl acetate under cooling in an ice bath. The excess solvent was decanted and the product concentrated under vacuum to afford 2-(((4*R*,6*R*)-6-methoxy-2,2-dimethyl tetrahydrofuro[3,4-*d*][1,3]dioxol-4-yl)methyl)isoindoline-1,3-dione **4** as an off-white crystalline solid (13 mg, 76%). The product exhibited spectral characteristics consistent with the ribosyl imide prepared by the methods of Ohrui and Kim [15,16].

### 3.13. Oxidation of 3-Hydroxy-2-(((4R,6R)-6-methoxy-2,2-dimethyltetrahydrofuro[3,4-d][1,3] dioxol-4-yl)methyl)isoindolin-1-one ***3*** with PCC/SiO_2_: Ribosyl Imide ***4***

To a solution of 3-hydroxy-2-(((4*R*,6*R*)-6-methoxy-2,2-dimethyltetrahydrofuro [3,4-*d*][1,3]dioxol-4-yl)methyl)isoindolin-1-one **3** and dichloromethane (5 mL) in a 25 mL round-bottom flask fitted with a magnetic stir bar was added a ground (see above) mixture of PCC (32.8 mg, 0.15 mmol) and silica gel (32.8 mg). The progress of the reaction was monitored at intervals by TLC (toluene/ethyl acetate, 1:1). Upon complete consumption of the starting material (8 h), the reaction mixture was diluted with ethyl acetate (5 mL) and filtered through a bed of Celite^®^ while washing with diethyl ether (3 × 5 mL). The filtrate was concentrated under vacuum to afford the product imide 2-(((4*R*,6*R*)-6-methoxy-2,2-dimethyltetrahydrofuro[3,4-*d*][1,3]dioxol-4-yl)methyl)isoindoline-1,3-dione **4** (10 mg, 59%) as an off-white crystalline solid. The product exhibited spectral characteristics consistent with the ribosyl imide prepared by the methods of Ohrui and Kim [15,16].

### 3.14. Oxidation of 3-Hydroxy-2-(((4R,6R)-6-methoxy-2,2-dimethyltetrahydrofuro[3,4-d][1,3] dioxol- 4-yl)methyl)isoindolin-1-one ***3*** with IBX: Ribosyl Imide ***4***

To a 25 mL round-bottom flask fitted with a stir bar was added acetonitrile (2.5 mL)*,* followed by 3-hydroxy-2-(((4*R*,6*R*)-6-methoxy-2,2-dimethyltetrahydrofuro[3,4-*d*][1,3] dioxol-4-yl)methyl)isoindolin-1-one **3** (17 mg, 0.050 mmol) and acetonitrile (2.5 mL). Stirring was commenced, and IBX (42 mg, 0.15 mmol) was added in one portion, followed by heating under reflux at 80 °C for 24 h (drying tube). The progress of the reaction was monitored by TLC (toluene/ethyl acetate, 2:1), whereupon full conversion of starting material **3** into the corresponding imide **4** was affected. The reaction mixture was filtered through a bed of Celite^®^, the filter cake was washed with ethyl acetate (3 × 5 mL), and the combined filtrate was concentrated. The crude product was triturated with a few milliliters of dichloromethane/pentane, followed by cooling (ice water bath). The excess solvent was decanted and the product concentrated under vacuum to afford the desired imide **4** as an off-white crystalline solid (12 mg, 71%). The spectral properties of the product were identical in all respects to the products of both the NiO_2_ and PCC/SiO_2_ oxidations.

## 4. Conclusions

We have introduced nickel peroxide as an oxidant for the conversion of hydroxylactams to imides. While there are isolated examples of nickel peroxide-mediated alcohol oxidation in general, the specific use of NiO_2_ in this hydroxylactam to imide conversion constitutes a unique application of this reagent preparation, and it compared favorably with two popular oxidation protocols, PCC and IBX. The NiO_2_ method proved useful for the oxidation of alcohol substrates, which are nitrogen bearing despite the occurrence of a lactam substrate and sensitive imide product as the nitrogen-bearing moiety. Similarly, the phthalimide-derived hydroxylactam moiety borne by the protected ribose scaffold responded well to NiO_2_ under standard conditions with the survival of all protecting groups. In terms of expense, the NiO_2_ method is clearly the most favorable, requiring only hypochlorite and nickel salt for preparation with nominal isolation issues. It should be mentioned that the toxicity issues with oxochromium (VI) reagents are known [38], so consequently, on a larger scale, the PCC/SiO_2_ method would be the least favorable. Similarly, the safety issues with the preparation of IBX should be considered, particularly the propensity of damp samples to detonate [39]. Given the environmental and sustainability issues associated with many oxo-metal reagent systems, catalytic methods utilizing ‘eco-friendly’ metals such as iron [40] are always welcome for evaluation and development and will be topics for further study in our laboratories.

## Data Availability

Not applicable.

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
