# Peer review of "An Oxidation Study of Phthalimide-Derived Hydroxylactams"

_molecules, 2022, doi:10.3390/molecules27020548_

Round 1
Reviewer 1 Report
In this work, the authors reported an oxidation study of phthalimide-based hydroxylactams. However, the work is not matural for publication at current status. The following issues need to be fully addressed before resubmision.
- For characterization of the new compound 3, FT-IR is missing; also the original NMR spectra and FT-IR spectra need to be provided.
- Why the freshly prepared NiO2 is necessary, the oxidant needs to be fully characterized to identify the difference with commercially available sources. Further, how to reuse the oxidant?
- How about the other metal oxides such as iron oxides? Can they replace the NiO2?
- Can the process be converted to a catalytic process?
Reviewer 2 Report
Figure 1 and Scheme 1 should be compiled to the single scheme.
Application of similar-type catalytic systems in the organic synthesis should be shortly discussed.
Mobile phases for TLC experiments should be specified.
Paragraph 3.11
- Signals in the IR spectrum should be assigned specific molecular segments.
- Signals in NMR spectra should be assigned specific nuclei
Round 2
Reviewer 1 Report
na